# The Impact of Anti-SARS-CoV-2 Vaccine in Patients with Systemic Lupus Erythematosus: A Multicentre Cohort Study

**DOI:** 10.3390/vaccines10050663

**Published:** 2022-04-22

**Authors:** Maria Gerosa, Tommaso Schioppo, Lorenza Maria Argolini, Savino Sciascia, Giuseppe Alvise Ramirez, Gabriella Moroni, Renato Alberto Sinico, Grazia Bonelli, Federico Alberici, Federica Mescia, Luca Moroni, Francesco Tamborini, Paolo Miraglia, Chiara Bellocchi, Lorenzo Beretta, Dario Roccatello, Lorenzo Dagna, Enrica Bozzolo, Roberto Caporali

**Affiliations:** 1Department of Clinical Sciences and Community Health, Research Center for Adult and Pediatric Rheumatic Diseases, University of Milan, 20122 Milan, Italy; maria.gerosa@unimi.it (M.G.); roberto.caporali@unimi.it (R.C.); 2ASST Pini CTO, Lupus Clinic, Division of Clinical Rheumatology, 20122 Milan, Italy; lorenza.argolini@hotmail.it; 3ASST Santi Paolo e Carlo, Medicina Generale I, 20142 Milan, Italy; 4Department of Clinical and Biological Sciences, University of Turin, 10124 Turin, Italy; savino.sciascia@unito.it; 5Nephrology and Dialysis Unit & CMID (Center of Research of Immunopathology and Rare Diseases), Coordinating Center of the Network for Rare Diseases of Piedmont and Aosta Valley, San Giovanni Bosco Hub Hospital of Turin, Department of Clinical and Biological Sciences, University of Turin, 10154 Turin, Italy; paolo.miraglia1@edu.unito.it (P.M.); dario.roccatello@unito.it (D.R.); 6IRCCS Ospedale San Raffaele, Unit of Immunology, Rheumatology, Allergy and Rare Diseases, 20132 Milan, Italy; ramirez.giuseppealvise@hsr.it (G.A.R.); moroni.luca@hsr.it (L.M.); dagna.lorenzo@unisr.it (L.D.); bozzolo.enrica@hsr.it (E.B.); 7Faculty of Medicine and Surgery, Università Vita-Salute San Raffaele, 20132 Milan, Italy; 8Department of Biomedical Sciences, Humanitas University, IRCCS Humanitas Research Hospital, 20072 Milan, Italy; gabriella.moroni@hunimed.eu; 9Renal Unit, Department of Medicine and Surgery, Università degli Studi di Milano Bicocca and ASST-Monza, 20900 Monza, Italy; renato.sinico@unimib.it (R.A.S.); g.bonelli2@campus.unimib.it (G.B.); 10Department of Medical and Surgical Specialties, Radiological Sciences and Public Health, University of Brescia, 25121 Brescia, Italy; federico.alberici@gmail.com (F.A.); federica.mescia@gmail.com (F.M.); 11Fondazione Ca’ Granda IRCCS Ospedale Maggiore Policlinico Milano, Divisione di Nefrologia e Dialisi, 20122 Milan, Italy; francesco_tamborini@asst-pavia.it; 12Fondazione IRCCS Ca’ Granda Ospedale Maggiore Policlinico di Milano, Referral Centre for Systemic Autoimmune Diseases, 20122 Milan, Italy; chiara.bellocchi@unimi.it (C.B.); lorberimm@hotmail.com (L.B.)

**Keywords:** systemic lupus erythematous, SARS-CoV-2 infection, SARS-CoV-2 vaccine, flare, side effect, immunisation

## Abstract

Vulnerable subjects, including systemic lupus erythematosus (SLE) patients, have been prioritised to receive anti-SARS-CoV-2 vaccines. Few data about the safety of these vaccines in SLE are available. The aim of our study is to investigate the safety of anti-SARS-CoV-2 vaccines in SLE. We included 452 SLE patients, referring to seven tertiary centres, who were immunised. A total of 119 (26%) reported side effects (SE) after the first and/or the second shot (the most frequent SE were fever, local reaction, fatigue, and arthralgia). Patients with constitutional symptoms and those on an immunosuppressive regimen (especially belimumab) showed more SE. In addition, 19 (4%) had a flare after the immunisation (flares classified by organ involvement: six musculoskeletal with constitutional symptoms, four renal, three cardio-respiratory, three haematological, two mucocutaneous). None of the patients needed hospitalisation and none died. Moreover, 15 required a transient increase in corticosteroids and four were treated with steroid pulses. One patient required an additional rituximab course. Anti-dsDNA, moderate/high DAS before vaccine, and belimumab were found more frequently in patients with disease flare. Anti-SARS-CoV-2 vaccines are safe in SLE patients, and they should be recommended in these patients, as the potential benefits widely outweigh the risk of SE. Treatment adjustment might be considered with the aim of minimising SE risk and flare.

## 1. Introduction

Severe acute respiratory syndrome coronavirus 2 (SARS-CoV-2) infection can cause serious respiratory complications including interstitial pneumonia and acute respiratory distress syndrome, frequently requiring intensive care and associated with elevated mortality rates [1].

Since the onset of the COVID-19 pandemic, great effort has been made to develop a vaccine against the SARS-CoV-2 virus, particularly for patients with a higher susceptibility to life-threatening severe complications.

Patients with inflammatory rheumatic diseases (RDs) are at greater risk of infections compared to the general population. Among RDs, patients with systemic lupus erythematosus (SLE) represent a unique population, prone to infections due to concomitant disease-related immune dysfunction, immunosuppressive therapy, and comorbidities [2]. In this clinical scenario, the use of vaccines represents one of the most effective strategies to limit infection-related morbidity and mortality. However, many questions have been raised about the effects of vaccines on the immune system and their potential role as a trigger for SLE onset or flare. Although a small study on 10 patients reported a temporal relationship between hepatitis B vaccine and the occurrence of SLE and another study showed a transient increase in anti-dsDNA antibody titres without disease flares after seasonal flu vaccination, to the best of our knowledge, none of the available vaccines have been recognised to be able to induce SLE [3,4,5]. Consequently, EULAR has recently highlighted the importance of immunisation against the most common vaccine-preventable diseases, advocating for an annual assessment of vaccine status in SLE patients [6]. 

Among the anti-SARS-CoV2 vaccines available on the market in Italy, Pfizer–BioNTech mRNA COVID-19 BNT162b2 and Moderna COVID-19 mRNA-1273 have been licensed and used both in Europe and North America, while AstraZeneca ChAdOx1-S was authorised in Europe only. Few data about the safety and efficacy of these vaccines are available in SLE patients, as these patients are generally excluded from all the phases of vaccine development [7]. 

A survey-based study reported that anti-SARS-CoV-2 vaccines were well tolerated in a large cohort of SLE patients, with a minimal rate of disease flare, in comparison to the annual rate of flare observed in the same population in the previous year [8]. Conversely, in a recent study on 100 SLE patients, there were 27 flares after the immunisation (9% and 20% after the first and the second dose, respectively). Seven patients presented flare after both doses [9]. 

In a recent review, the benefits of the anti-SARS-CoV-2 vaccine far outweighed the potential risks of side effects or disease flares [10].

The aim of our study is to investigate the safety of different anti-SARS-CoV-2 vaccines in SLE patients in a multicentre cohort of tertiary hospitals of Northern Italy.

## 2. Material and Methods

### 2.1. Study Design

The study was designed as a retrospective, observational, multicentre cohort study. 

### 2.2. Setting

Data regarding SLE patients who had been administered the anti-SARS-CoV-2 vaccine between December 2020 and October 2021 were retrospectively collected. The patients included in the study referred to seven SLE tertiary centres: Lupus Clinic, Clinical Rheumatology Division of ASST Pini-CTO, Milan; CMID-Nephrology Unit of Ospedale Giovanni Bosco, Turin; IRCCS Humanitas Research Hospital; Renal and Rheumatology Units, San Gerardo Hospital, Monza; ASST Spedali Civili; Lupus Clinic at Unit of Immunology, Rheumatology, Allergy and Rare Diseases, IRCCS Ospedale San Raffaele, Milan; Referral Center for Systemic Autoimmune Diseases, Fondazione IRCCS Ca’ Granda Ospedale Maggiore Policlinico, Milan. The analysis is part of a study to collect observational data from SLE patients and it was approved by the Ethics Committee Comitato Etico Milano Area 2 (approval no. 0002450/2020). The study was conducted conforming to the Declaration of Helsinki.

### 2.3. Participants

All consecutive adult patients (≥18 years of age) referring to participant centres with a previous diagnosis of SLE who had received anti-SARS-CoV-2 vaccines between 29 December 2020 and 31 October 2021 (BNT162b2 mRNA COVID-19 vaccine by Pfizer or mRNA-1273 by Moderna Biotech or ChAdOx1-S by AstraZeneca) were included in the study. All patients provided written informed consent. The diagnosis of SLE was made in accordance with the 1997 ACR classification criteria or the 2019 EULAR–ACR criteria [11,12]. 

### 2.4. Variables

Data regarding demographics and clinical and serological features of SLE (including organ involvement, disease duration, autoimmune profile, ongoing treatment, disease activity before and after anti-SARS-CoV-2 vaccine, possible treatment modifications related to vaccination and comorbidities) were collected and gathered in a database for statistical analysis. A history of confirmed SARS-CoV-2 infection before and/or after immunisation was recorded. Moreover, when available, laboratory parameters (CRP, ESR, gamma globulins) were collected at the last follow-up visit before and at the first follow-up visit after the vaccine. Disease activity was assessed according to the British Isles Lupus Assessment Group (BILAG) 2004 index [13]. Disease flare was defined as an increase of at least 4 points in SLEDAI-2K score, or a new “A” or “B” score at BILAG in at least one domain. Disease manifestations were classified as “flare” when they were persistent, in contrast to “side effects” when manifestations were transient and self-limited.

### 2.5. Statistical Analysis

Descriptive statistics were used to summarise patients’ demographic and clinical data by using median, interquartile range (IQR), absolute numbers, and percentages. All these variables were then investigated as risk factors of the following outcomes: side effects and SLE flare(s) after SARS-CoV-2 vaccination. The comparisons of continuous variables between groups of patients were assessed through the Mann–Whitney nonparametric test. The association between categorical variables was assessed by performing Fisher’s exact test. A *p*-value < 0.05 was considered significant. All analyses were performed using R software version 3.5.2 (R Foundation for Statistical Computing, Vienna, Austria) with package Rcmdr (version 2.5-1).

## 3. Results

Four hundred fifty-two patients, who had received the anti-SARS-CoV-2 vaccine, were included in the study. Demographic, comorbidity, and SLE disease characteristics and therapies before SARS-CoV-2 vaccination are reported in Table 1. At the last visit, 12 (3%) were not receiving therapy, 322 (71%) were on prednisone at a low–medium dose, 374 (83%) were taking hydroxychloroquine, and 228 (50%) were being treated with an immunosuppressive drug (Table 1). Only nine patients had transiently discontinued immunosuppressants before vaccination. At the time of first vaccine injection, almost 90% of patients were in remission or low disease activity according to the Low Disease Activity State (LLDAS) [14]. 

Most patients (411, 90.93%) received the BNT162b2 mRNA COVID-19 vaccine by Pfizer, 34 (7.52%) received mRNA-1273 by Moderna Biotech, and the remaining patients (7) received ChAdOx1-S by AstraZeneca. Thirteen patients had a confirmed SARS-CoV-2 infection before immunisation. Infections occurred with a median time of 6.5 months before immunisation (interquartile range: 4.5–10.75).

As reported in Table 2, 119 (26.33%) patients reported adverse symptoms after the first or second shot. Fifty-six (12.38%) showed side effects after the first and 96 (20.80%) after the second shot. Among side effects, the most frequent were fever, pain, swelling and redness at the site of injection, fatigue, and arthralgia. 

According to data reported in Table 3, patients with constitutional symptoms and those treated with at least one immunosuppressant (especially those on belimumab) showed an increased rate of side effects. Conversely, patients with no therapy before the vaccine had a reduced risk of adverse events. There was no association between the history of infection before vaccination and the risk of adverse reactions. Seventy-seven (17%) of patients experienced SARS-CoV-2 infection after vaccination, with a median time to infection of 7 months (IQR 6–8 months).

There was no significant difference in the percentage of active BILAG domains before and after vaccination (Table 4). Nineteen (4%) patients flared up after immunisation, with a median time to relapse of 7 ± 2 days. The incidence rate of flare after immunisation was not superior to that observed in the same cohort during the period between 2015 and 2019.

The differential distribution of baseline characteristics among patients with vs. without flares after SARS-CoV-2 vaccination is reported in Table 5. 

Anti-dsDNA positivity, moderate or high DAS before vaccine, and the use of belimumab were significantly more frequent in patients who experienced a flare. Moreover, patients who experienced a flare after SARS-CoV-2 vaccination displayed a significantly higher rate of adverse events after vaccination: 63% of those who had a flare also had an adverse reaction to the vaccination, while only 25% of those who did not have a flare had an adverse reaction. 

Among these 19 patients, 84.2% were on hydroxychloroquine, 78.9% were taking low-dose prednisone, 77.3% were being treated with immunosuppressants (mycophenolate 42.1%, belimumab 36.8%, methotrexate 5.1%, azathioprine 5.1%, and cyclophosphamide 5.1%). None of these patients discontinued therapy before the vaccination. 

None of the patients experiencing a flare had a history of SARS-CoV-2 infection before vaccination.

Qualitatively, six (31.6%) patients experienced a musculoskeletal flare associated with constitutional symptoms (one patient experienced constitutional symptoms only and one patient musculoskeletal symptoms only), four (21%) patients had a renal flare, three (16%) had cardio-respiratory manifestations, three (16%) haematological, and two (10%) a mucocutaneous flare.

In 79% (15) of the cases, the flare was treated with a transient increase in oral steroid dosage, while in four cases, intravenous steroid pulses were required. Moreover, in one case, a rituximab course was repeated in a patient who had received the first course nine months before. None of the patients needed to be hospitalised and no case of death was recorded.

Moreover, anti-dsDNA antibodies became positive in 36 patients, although this finding was not associated with a higher disease flare rate. 

## 4. Discussion

Our study explored the safety of anti-SARS-CoV-2 vaccination in a large multicentric cohort of SLE patients from Northern Italy. Our data showed that anti-SARS-CoV-2 vaccines are overall safe in SLE patients as we recorded a very low rate of side effects and none of the side effects were severe or required hospitalisation. The number of patients experiencing moderate to severe local or systemic adverse reactions in our study is lower than previous studies, both including RD patients and the healthy population [15]. 

In our cohort, the most frequent post-injection systemic manifestations were fever, arthralgia and fatigue, both after the first and the second boost. Compared to other studies, all these symptoms were less frequent than previously described (in few studies evaluating SLE patients separately, fatigue and headache were reported by 30–70% of patients) [9,10,16,17,18]. Interestingly, fever was much less frequent (15–20%), but still considerably more than what we observed (4–8% after the first and second boost). Diverging methodological approaches might account for this finding. In previously available studies, local and systemic reactions were self-reported surveys or questionnaires. The data included in our study were collected during outpatient visits with an accurate oral face-to-face interview performed by a trained physician. Moreover, because our data were collected over a longer period of time than the other studies, patients who received immunisation more recently could have been more reassured by clinicians, who were supported by the encouraging literature and who tended to pay less attention to non-specific symptoms (disease flare vs. side effects after the vaccine). 

The presence of constitutional symptoms, the absence of therapy, and the use of at least one immunosuppressant, particularly belimumab, before vaccination were factors found to be significantly associated with a higher risk of having side effects after immunisation. To our knowledge, this is the first study evaluating this aspect in SLE patients. We can speculate that patients with constitutional symptoms are more likely to experience a worsening after vaccination. The association with the use of immunosuppressants can suggest that a more severe disease, requiring a disease-modifying drug (DMARDs), could facilitate the occurrence of side effects. The identification of belimumab, as the most implicated therapy, is in line with this hypothesis as more than 50% of patients were receiving this therapy in addition to conventional treatments, probably identifying a subgroup of patients with a more aggressive disease (in Italy, due to eligibility criteria, belimumab can be prescribed only in patients with an active disease despite the conventional immunosuppressive therapy). The lack of any association with the serological profile is reassuring: our data confirmed that positive anti-phospholipid antibodies do not represent a major concern for the risk of thrombosis in patients undergoing anti-SARS-CoV-2 vaccination [19]. 

Another important safety issue regards the effects of vaccination on disease activity. The potential of vaccines and adjuvants to trigger SLE reactivation has been a matter of concern for many years. Less is known about the impact of novel vaccine technologies, such as liposomal RNA, in triggering inflammation. However, recent studies on different types of live-attenuated or inactivated vaccines, including papilloma virus and influenza vaccines, did not report an increased risk of flare and vaccinations are currently recommended, also by EULAR, in these patients [20]. In our study, the incidence of a new disease flare after the first and/or the second dose was very low (not superior when compared to the period 2015–2019): reported manifestations were mainly involving the musculoskeletal, renal, and cutaneous domains. Mean BILAG scores did not change over time and disease flares were usually mild or moderate and none of them required hospitalisation. These observations are in line with previous studies, both considering patients with SLE and other RDs. Most studies reported a rate of flare ranging from 3% to 12%, with a preponderance of musculoskeletal and constitutional involvement [21,22,23,24]. Interestingly, patients, who experienced an adverse reaction, also had a higher disease flare rate. However, caution is warranted in interpreting these data, as some of the most frequent adverse events can mimic SLE clinical manifestations. In this regard, fever, arthralgias, or mild arthritis could have been misinterpreted and attributed to both an adverse reaction to immunisation and to SLE. We can also assume that some side effects of vaccination can trigger disease flares. In line with this assumption, we observed a case of renal flare after metabolic acidosis induced by diarrhoea. Conversely, our data also suggest that the prompt treatment of adverse events might potentially prevent further complications in the disease course. 

Risk factors for SLE reactivation were positive anti-dsDNA, moderate/high disease activity before vaccination, and the use of belimumab. This is not surprising, as SLE patients with high disease activity and severity have been identified as risk factors for flares in different disease settings [25,26] and constitute the reference target for belimumab treatment. Moreover, this finding is in line with a previous study that identified a history of at least one flare during the past year, an indirect indicator of unstable disease, as the most relevant risk factor for further flares [8]. 

The lack of immunogenicity and efficacy data constitutes a major limitation of our study. We also did not obtain experimental data, which could have offered potential hints to understand immune dynamics in patients with SLE challenged with either anti-SARS-CoV-2 vaccines or wild-type infection. In particular, data about interferon alpha responses, which appear dysregulated both in SARS-CoV-2 infection and SLE, are warranted for a comprehensive understanding of the immunological profile of patients with connective tissue disorders facing SARS-CoV-2 infection. Moreover, almost all the included patients were Caucasian and, therefore, our data should be replicated in other populations. Conversely, the strength of our study lies in the large number of clinically and serologically well-characterised SLE patients, regularly followed in seven tertiary referral centres. 

## 5. Conclusions

Our data are reassuring and confirms that anti-SARS-CoV-2 vaccination is overall safe in SLE patients. Immunisation against SARS-CoV-2, according to the EULAR and ACR recommendations, should be recommended in this clinical setting since the potential benefits widely outweigh the risk of adverse events. Therapy adjustment could be required to minimise the risk of side effects and disease flare, consequently ensuring satisfying protection against SARS-CoV-2 infection. Our work suggests to carefully follow up with SLE patients in the weeks following active immunisation to promptly identify disease flares and to start adequate treatment. 

## Figures and Tables

**Table 1 vaccines-10-00663-t001:** Demographic, comorbidity, and SLE disease characteristics and therapies before SARS-CoV-2 vaccination.

Demographics	
Age, years, median (IQR)	48 (35–56)
Disease duration, months, median (IQR)	128 (73,75–197,25)
Female, n (%)	417 (92.25)
**Laboratory, median (25th–75th)**	
ESR, mm/h ^+^	13 (7–21)
CRP, mg/dL	0.5 (0.29–0.51)
IgG, mg/dL ^£^	1138 (882.5–1405)
**Serology and organ involvement ever, n (%)**	
Anti-dsDNA positivity ^§^	382 (85)
Anti-Sm positivity ^$^	48 (11)
Low complement (C3 and/or C4)	159 (35)
Musculoskeletal involvement	382 (85)
Mucocutaneus involvement	294 (65)
Renal involvement	224 (50)
Urinary abnormalities	84 (19)
NPSLE involvement	46 (10)
Cardiopulmonary involvement	93 (21)
Haematological involvement	149 (33)
Constitutional symptoms	158 (35)
Gastrointestinal involvement	16 (4)
Ophthalmic involvement	12 (3)
Anti-phospholipid antibodies positivity	142 (31)
Secondary anti-phospholipid syndrome	48 (11)
**Therapy, n (%)**	
No therapy	12 (3)
Hydroxychloroquine	374 (83)
Prednisone	322 (71)
Immunosuppressor *	228 (50)
Cyclophosphamide ever	112 (25)
Rituximab ever	59 (13)
**Disease activity before SARS-CoV-2 vaccination, n (%)**	
Remission/LLDAS	402 (89)
Moderate/High DAS	50 (11)
**SARS-CoV-2 infection, n (%)**	
Infection before vaccination	13 (3)
Infection after vaccination	77 (17)

^+^ not available for 3 patients. ^£^ not available for 286 patients. ^§^ not available for 17 patients. ^$^ not available for 1 patient. * Azathioprine, mycophenolate, belimumab, cyclophosphamide, leflunomide, adalimumab, rituximab, tocilizumab, tacrolimus. IQR: interquartile range. ESR: erythrocyte sedimentation rate. CRP: C-reactive protein. NPSLE: neuropsychiatric involvement in systemic lupus erythematosus. LLDAS: lupus low disease activity score, DAS: disease activity score.

**Table 2 vaccines-10-00663-t002:** Side effects after vaccination according to side effects due to SARS-CoV-2 vaccination.

After first shot, n (%): 56 (12.38)	
Headache	7
Arthralgia	12
Myalgia	7
Fever	19
Local reaction	17
Gastrointestinal symptoms	6
Lymphadenopathy	2
Fatigue	12
Rash	3
Other	2
**After second shot, n (%): 94 (20.80)**	
Headache	6
Arthralgia	28
Arthritis	1
Myalgia	9
Fever	36
Local reaction	17
Gastrointestinal symptoms	11
Lymphadenopathy	3
Fatigue	20
Rash	3
**After first or second shot, n (%): 119 (26.33)**	

**Table 3 vaccines-10-00663-t003:** Distribution of demographic and SLE patient characteristics according to side effects due to SARS-CoV-2 vaccination.

	Side Effects after First or Second Shot (n = 119)	No Side Effects after First or Second Shot (n = 333)	*p*-Value (<0.05)
Age, years, median (IQR)	46 (33.5–54)	48 (35.75–57)	0.1817
Disease duration, months, median (IQR)	138 (76–262.5)	126 (73–193)	0.3016
Musculoskeletal involvement, % (n)	84.9 (101)	84.4 (281)	1.00
Mucocutaneus involvement, % (n)	71.4 (85)	62.8 (205)	0.09403
Renal involvement, % (n)	42.0 (50)	52.3 (174)	0.06917
NPSLE, % (n)	13.4 (16)	9 (30)	0.2151
Cardiopulmonary involvement, % (n)	22.7 (27)	19.8 (66)	0.5109
Haematological involvement, % (n)	32.8 (39)	33 (110)	1.00
Constitutional symptoms, % (n)	48.7 (58)	30 (100)	0.000325 *
Gastrointestinal involvement, % (n)	4.2 (5)	3.3 (11)	0.7726
Ophthalmic involvement, % (n)	0.8 (1)	3.3 (11)	0.1973
Secondary anti-phospholipid syndrome, % (n)	10.9 (13)	10.5 (35)	0.8641
Anti-phospholipid antibodies positivity, % (n)	26.2 (31)	33.6 (112)	0.1366
Anti-dsDNA positivity, % (n)	30.7 (35)	27.4 (88)	0.5453
Anti-Sm positivity, % (n)	12.6 (15)	9.9 (33)	0.4881
Low complement (C3 and/or C4), % (n)	37.8 (45)	34.2 (114)	0.5033
ESR, mm/h, median (IQR)	14 (7.25–19.75)	13 (7–22)	0.7303
CRP, mg/dL, median (IQR)	0.5 (0.1–0.5)	0.5 (0.3–0.6)	0.3122
Presence of urinary abnormalities, % (n)	9.2 (11)	21.9 (73)	0.002311
Moderate or high DAS before vaccine, % (n)	16 (19)	9.3 (31)	0.06011
No therapy before vaccine, % (n)	0 (0)	3.6 (12)	0.0419 *
At least 1 immunosuppressant ^#^, % (n)	63 (75)	46.8 (156)	0.002751 *
Mycophenolate, % (n)	31.9 (38)	23.1 (77)	0.06606
Methotrexate, % (n)	5.9 (7)	6.6 (22)	1.00
Belimumab, % (n)	21.8 (26)	13.5 (45)	0.03956 *
Rituximab ever, % (n)	11.8 (14)	13.5 (45)	0.7515
Prednisone therapy, % (n)	74.8 (89)	70 (233)	0.3468

* *p*-Value < 0.05. ^#^ Azatioprine, mycophenolate, belimumab, cyclophosphamide, leflunomide, adalimumab, rituximab, tocilizumab, tacrolimus. IQR: interquartile range. ESR: erythrocyte sedimentation rate. CRP: C-reactive protein. NPSLE: neuropsychiatric involvement in systemic lupus erythematosus. DAS: disease activity score.

**Table 4 vaccines-10-00663-t004:** BILAG score before and after SARS-CoV-2 vaccination.

BILAG before SARS-CoV-2 Vaccination, n (%)					
	A	B	C	D	E
Musculoskeletal	1 (0)	13 (3)	22 (5)	355 (79)	61 (13)
Mucocutaneus	0 (0)	6 (1)	11 (2)	270 (60)	165 (37)
Renal	7 (2)	41 (9)	15 (3)	157 (35)	232 (51)
NPSLE	1 (0)	3 (1)	3 (1)	42 (9)	403 (89)
Cardiopulmonary	1 (0)	1 (0)	2 (0)	71 (16)	377 (83)
Haematological	0 (0)	2 (0)	27 (6)	125 (28)	298 (66)
Constitutional	0 (0)	2 (0)	5 (1)	150 (33)	295 (65)
Gastrointestinal	0 (0)	1 (0)	0 (0)	18 (4)	433 (96)
Ophthalmic	0 (0)	0 (0)	0 (0)	14 (3)	438 (97)
**BILAG after SARS-CoV-2 Vaccination, n (%)**					
	**A**	**B**	**C**	**D**	**E**
Musculoskeletal	3 (1)	11 (2)	17 (4)	340 (75)	81 (18)
Mucocutaneus	0 (0)	5 (1)	11 (2)	256 (57)	180 (40)
Renal	10 (2)	35 (8)	14 (3)	141 (31)	252 (56)
NPSLE	0 (0)	3 (1)	3 (1)	38 (8)	408 (90)
Cardiopulmonary	0 (0)	3 (1)	2 (0)	63 (14)	384 (85)
Haematological *	1 (0)	2 (0)	26 (6)	112 (25)	308 (68)
Constitutional	3 (1)	5 (1)	6 (1)	143 (32)	295 (65)
Gastrointestinal	0 (0)	2 (0)	0 (0)	16 (4)	434 (96)
Ophthalmic	0 (0)	0 (0)	0 (0)	11 (2)	441 (98)

* Not available for 3 patients. NPSLE: neuropsychiatric involvement in systemic lupus erythematosus.

**Table 5 vaccines-10-00663-t005:** Distribution of baseline demographic and SLE patient characteristics according to disease flare after SARS-CoV-2 vaccination.

	Disease Flare after Vaccine (n = 19)	No Disease Flare after Vaccine (n = 430)	*p*-Value (<0.05)
Age, years, median (IQR)	52 (39.5–56.0)	48 (35.0–56.9)	0.8487
Disease duration, months, median (IQR)	144 (122–242.5)	127 (73–195)	0.2489
Musculoskeletal involvement, % (n)	78.9 (15)	84.8 (367)	0.5142
Mucocutaneus involvement, % (n)	57.9 (11)	64.5 (283)	0.6237
Renal involvement, % (n)	52.6 (10)	49.4 (214)	0.8187
NPSLE, % (n)	5.3 (1)	10.4 (45)	0.708
Cardiopulmonary involvement, % (n)	26.3 (5)	20.3 (88)	0.5617
Haematological involvement, % (n)	42.1 (8)	32.6 (141)	0.4554
Constitutional symptoms, % (n)	26.3 (5)	35.3 (153)	0.473
Gastrointestinal involvement, % (n)	5.3 (1)	3.5 (15)	0.5029
Ophthalmic involvement, % (n)	0 (0)	2.8 (12)	1.00
Secondary anti-phospholipid syndrome, % (n)	5.3 (1)	10.9 (47)	0.7079
Anti-phospholipid antibodies positivity, % (n)	26.3 (5)	31.9 (138)	0.802
Anti-dsDNA positivity, % (n)	55.6 (10)	27.1 (113)	0.01419 *
Anti-Sm positivity, % (n)	15.8 (3)	10.4 (45)	0.4414
Low complement (C3 and/or C4), % (n)	52.6 (10)	34.2 (149)	0.1391
ESR, mm/h, median (IQR)	19 (10–24.5)	13 (7–21)	0.1247
CRP, mg/dL, median (IQR)	0.42 (0.135–0.50)	0.50 (0.30–0.53)	0.4638
Presence of urinary abnormalities, % (n)	21.1 (4)	18.5 (80)	0.7643
Moderate or high DAS before vaccine, % (n)	26.3 (5)	10.4 (45)	0.04739 *
No therapy before vaccine, % (n)	0 (0)	2.8 (12)	1.00
At least 1 immunosuppressant ^#^, % (n)	73.7 (14)	50.1 (217)	0.0593
Mycophenolate, % (n)	42.1 (8)	24.7 (107)	0.106
Methotrexate, % (n)	5.3 (1)	6.5 (28)	1.00
Belimumab, % (n)	36.8 (7)	14.8 (64)	0.01838 *
Rituximab ever, % (n)	5.3 (1)	13.4 (58)	0.4904
Prednisone therapy, % (n)	78.9 (15)	70.9 (307)	0.6069
Side effects after vaccination, % (n)	63.2 (12)	24.7 (107)	0.00062 *

* *p*-Value < 0.05. ^#^ Azatiopryne, mycophenolate, belimumab, cyclophosphamide, leflunomide, adalimumab, rituximab, tocilizumab, IMP, tacrolimus. IQR: interquartile range. ESR: erythrocyte sedimentation rate. CRP: C-reactive protein. NPSLE: neuropsychiatric involvement in systemic lupus erythematosus. DAS: disease activity score.

## Data Availability

The data presented in this study are available on request from the corresponding author. The data are not publicly available due to privacy reasons.

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
