# Peer review of "The Impact of Anti-SARS-CoV-2 Vaccine in Patients with Systemic Lupus Erythematosus: A Multicentre Cohort Study"

_vaccines, 2022, doi:10.3390/vaccines10050663_

Round 1

Reviewer 1 Report

Gerosa et al., examined the safety of anti-SARS-CoV-2 vaccination in patients diagnosed with SLE in Northern Italy. SLE patients generally experienced very few side effects from the anti-SARS-CoV-2 vaccines, and none of these effects were severe enough to necessitate hospitalization. As a result of this study, authors claim that the proportion of patients experiencing moderate to severe local or systemic adverse reactions was lower than previous studies, which included RDs patients and the general population. 

I wonder why the authors include only 35-56 aged people. Why not the age group of below 35 or elderly. Besides, there is a lot of evidence pointing to the importance of type I IFNs in the course of COVID-19. It is well known that type I IFNs have antiviral properties, but many viruses have developed strategies to evade their inhibitory effects. It is also noteworthy that most patients with SLE have elevated levels of type I IFNs or have overexpressed type I IFN genes in circulating immune cells; therefore, IFNs may play a critical role in the convergence between SLE and COVID-19. The authors should have considered checking type I IFNs. 

SLE patients are at high risk of viral infections followed by bacteria, pneumonia, and death. I believe more research is needed in different SLE ethnic groups before claiming that COVID vaccines are safe for SLE patients. However, the authors have already highlighted it in conclusion. I would suggest elaborating it more. 

Author Response

Gerosa et al., examined the safety of anti-SARS-CoV-2 vaccination in patients diagnosed with SLE in Northern Italy. SLE patients generally experienced very few side effects from the anti-SARS-CoV-2 vaccines, and none of these effects were severe enough to necessitate hospitalization. As a result of this study, authors claim that the proportion of patients experiencing moderate to severe local or systemic adverse reactions was lower than previous studies, which included RDs patients and the general population.

  • Answer

We thank the Reviewer for his/her work and his/her comments.

I wonder why the authors include only 35-56 aged people. Why not the age group of below 35 or elderly. Besides, there is a lot of evidence pointing to the importance of type I IFNs in the course of COVID-19. It is well known that type I IFNs have antiviral properties, but many viruses have developed strategies to evade their inhibitory effects. It is also noteworthy that most patients with SLE have elevated levels of type I IFNs or have overexpressed type I IFN genes in circulating immune cells; therefore, IFNs may play a critical role in the convergence between SLE and COVID-19. The authors should have considered checking type I IFNs.

  • Answer

Included patients median age was 48 with a IQR (interquartile range) between 35-56, this mean that 25% of patients was under 35 years and 25% older than 56 years (see table 1). As reported in methods, only patients older than 18 years were included in the study. The youngest patient was 20-year-old and the oldest 90-year-old (these data are not reported in the text).

We do agree that IFN signature plays a pivotal role in SLE patients but assessment of interferon responses in relation to vaccination and/or previous infection was beyond the scopes of this observational real-life study focused on clinical data. We have included the potential role of IFN in this context within the discussion.

SLE patients are at high risk of viral infections followed by bacteria, pneumonia, and death. I believe more research is needed in different SLE ethnic groups before claiming that COVID vaccines are safe for SLE patients. However, the authors have already highlighted it in conclusion. I would suggest elaborating it more.

  • Answer

We thank the Reviewer for this comment. We added a sentence in the limitations of the present study since most of the included patients were Caucasians.

Reviewer 2 Report

Gerosa and colleagues examined the impact of SARS-CoV-2 vaccination on patients with SLE.  They compared the demographics of patients who experienced side effects after vaccination to patients who experienced no side effects.  They made similar comparisons between patients who experience flares and patients who did not experience flares.  They generally observed low rates of side effects, and their observations are consistent with vaccination against SARS-CoV-2 being safe in SLE patients.

Major Comments

1.  A major limitation of this study is that the authors are drawing conclusions about the safety of the vaccine in a patient population with SLE but they do not have a control population of individuals without SLE. Because of this limitation, it’s impossible to determine if the rate of side effects is higher or lower in SLE patients.  If medical records are available for study participants, the authors could reference the period before immunization to determine if the rate of flares increased following immunization.  This sort of historical control would strengthen the paper.

2.  Table 3 and Table 5 are difficult to keep straight. Table 3 says that anti-dsDNA antibodies were not associated with side effects and Table 5 says that anti-dsDNA antibodies were associated with flare.  I understand that these are potentially separate issues, but if one of the questions in the manuscript is whether or not vaccination triggers flares, then one could argue that a flare is a side effect of vaccination.  I don't know that this issue can be "fixed" but I think it is worth adding a few lines of text to directly address it.

3. If available, please include history of confirmed SARS-CoV-2 infection in the tables. A primary immunization in a person without a history of infection is immunologically very different from an immunization in a person who was previously infected.

Minor Comments

Line 31:  Being the sentence “The aim of our study…”

Line 139:  capitalize “table”

Line 153:  insert “received” immediately following “(7.52%)”

Line 222-223:  Revise “…our data were collected over a longer period…”

Author Response

Gerosa and colleagues examined the impact of SARS-CoV-2 vaccination on patients with SLE.  They compared the demographics of patients who experienced side effects after vaccination to patients who experienced no side effects.  They made similar comparisons between patients who experience flares and patients who did not experience flares. They generally observed low rates of side effects, and their observations are consistent with vaccination against SARS-CoV-2 being safe in SLE patients.

  • Answer

We thank the Reviewer for her/his work and comments.

Major Comments

  1. A major limitation of this study is that the authors are drawing conclusions about the safety of the vaccine in a patient population with SLE but they do not have a control population of individuals without SLE. Because of this limitation, it’s impossible to determine if the rate of side effects is higher or lower in SLE patients. If medical records are available for study participants, the authors could reference the period before immunization to determine if the rate of flares increased following immunization. This sort of historical control would strengthen the paper.
  • Answer

We thank the Reviewer for this comment. We collected historical data in the same cohort for the period  between 2015 and 2019. We did not observed a higher flare incidence rate after the vaccine when compared to the period 2015-2019. We add the data in the results.

  1. Table 3 and Table 5 are difficult to keep straight. Table 3 says that anti-dsDNA antibodies were not associated with side effects and Table 5 says that anti-dsDNA antibodies were associated with flare. I understand that these are potentially separate issues, but if one of the questions in the manuscript is whether or not vaccination triggers flares, then one could argue that a flare is a side effect of vaccination. I don't know that this issue can be "fixed" but I think it is worth adding a few lines of text to directly address it.
  • Answer

We thank the Reviewer for having pointed out this interesting topic, which indeed is part of a large and unresolved debate. Patients with immune-mediated disorders were not part of registration trials with anti-SARS-CoV-2 vaccines, thus preventing the availability of high-quality, unbiased evidence on the clinical characteristics of post-vaccine events in this population. Observational cohort studies in patients with rheumatic disorders have used different definitions of flares and post-vaccine adverse events. Some Authors chose to apply a broad definition of flare to include any patient-reported worsening in disease-related symptoms, such as arthralgia, or the need of symptomatic therapy (eg. Barbhaiya M et al., Ann Rheum Dis, 2021 or Cherian S et al., Rheumatol Int, 2021) while others relied on Physician’s impression (Furer V et al, Ann Rheum Dis, 2021 or Felten R et al., Lancet Rheumatol, 2021). Most evidence converges towards the fact that inflammatory manifestations short after the vaccination tend to spontaneously subside, which suggests that they do not associate with major changes in disease pathophysiology and do not need major therapy modification. Therefore, we decided to rely on the concept of sustained activity and persistence of symptoms to support a definition of proper disease flare, mirroring what usually happens in routine rheumatology practice besides vaccination. This consideration has now been highlighted in the methods.

  1. If available, please include history of confirmed SARS-CoV-2 infection in the tables. A primary immunization in a person without a history of infection is immunologically very different from an immunization in a person who was previously infected.
  • Answer

We thank the Reviewer for this helpful suggestion. Thirteen patents (3%) had a history of COVID-19 before vaccination. We have now included this finding in the text. There was no association between a history of COVID-19 before vaccination and the occurrence of disease flares after vaccination.

Minor Comments

Line 31:  Being the sentence “The aim of our study…”

Line 139:  capitalize “table”

Line 153:  insert “received” immediately following “(7.52%)”

Line 222-223:  Revise “…our data were collected over a longer period…”

  • Answer

We addressed all the minor comments. We thank the Reviewer for these corrections.

Reviewer 3 Report

My impression is this is a very important paper.
This paper includes two topics: the adverse effect of vaccination itself and post-vaccination SLE flares.

This manuscript is missing the following two very important explanations.
1. The authors describe that the patients who showed lupus flare after vaccination were often positive anti-ds-DNA antibodies at baseline. In table 5, do you mean that the antibody negative cases are patients who originally positive but turned negative as a result of various treatments? Or was the case originally negative for anti-ds-DNA antibodies?
Do you have any data on titer of anti-ds-DNA antibodies? 
The titer is a very important point when considering the next issue.

2. The decision that Belimumab users are prone to flare is poorly considered.
The author states that flares are more frequent in patients who use belimumab because they have a "more aggressive disease".(line 233) Is there any other inference?
Can you provide data on anti-ds-DNA antibody titers, complement levels, urinary protein, or BILAG that would indicate "more aggressive disease" in the belimumab group?

3. In Table 5, the authors state that patients who had a post-vaccination SIDE EFFECT have a higher rate of SLE flares, but I have not seen sufficient discussion of this point in the discussion paragraph. The authors assume an effect of adjuvants (line 241), but is this sufficient?

In the disucussion paragraph, there is a lot of rewriting of the content of the RESULT. Please remove these and add a little more discussion content.

Author Response

My impression is this is a very important paper. This paper includes two topics: the adverse effect of vaccination itself and post-vaccination SLE flares.

  • Answer

We thank the Reviewer for his/her work.

This manuscript is missing the following two very important explanations.

  1. The authors describe that the patients who showed lupus flare after vaccination were often positive anti-ds-DNA antibodies at baseline. In table 5, do you mean that the antibody negative cases are patients who originally positive but turned negative as a result of various treatments? Or was the case originally negative for anti-ds-DNA antibodies? Do you have any data on titer of anti-ds-DNA antibodies? The titer is a very important point when considering the next issue.
  • Answer

We thank the Reviewer for this comment. In Italy, belimumab can be only prescribed in a certain subset of SLE patients (patients not completely responding to  conventional therapy, with active disease -SKEDAI>10- or with serological activity anti-DNA medium-high titer and/or low complement). We added this information in the discussion.

In table 5, we have considered the positivity to anti-dsDNA antibodies at baseline: even if some of these patients have turned negative at the time of vaccination as a result of various previous treatments, they have been included in the anti-dsDNA positive group due to their previous positivity.  As it was a real-life multicenter study, we did not collect anti-dsDNA antibody titers of all patients but categorized positive only those patients displaying a medium/high antibody titer. 

  1. The decision that Belimumab users are prone to flare is poorly considered.

The author states that flares are more frequent in patients who use belimumab because they have a "more aggressive disease" (line 233) Is there any other inference? Can you provide data on anti-ds-DNA antibody titers, complement levels, urinary protein, or BILAG that would indicate "more aggressive disease" in the belimumab group?

  • Answer:

According to local regulation, belimumab can be used only in SLE patients who have an active disease despite the conventional immunosuppressive therapy. This created a selection bias that can be hardly controlled. We added this information in the discussion.

  1. In Table 5, the authors state that patients who had a post-vaccination SIDE EFFECT have a higher rate of SLE flares, but I have not seen sufficient discussion of this point in the discussion paragraph. The authors assume an effect of adjuvants (line 241), but is this sufficient?
  • Answer

We thank the Reviewer for having pointed out this interesting topic, which indeed is part of a large and unresolved debate. Patients with immune-mediated disorders were not part of registration trials with anti-SARS-CoV-2 vaccines, thus preventing the availability of high-quality, unbiased evidence on the clinical characteristics of post-vaccine events in this population. Observational cohort studies in patients with rheumatic disorders have used differnt definitions of flares and post-vaccine adverse events. Some authors chose to apply a broad definition of flare to include any patient-reported worsening in disease-related symptoms, such as arthralgia, or the need of symptomatic therapy (eg. Barbhaiya M et al., Ann Rheum Dis, 2021 or Cherian S et al., Rheumatol Int, 2021) while others relied on Physician’s impression (Furer V et al, Ann Rheum Dis, 2021 or Felten R et al., Lancet Rheumatol, 2021). Most evidence converges towards the fact that inflammatory manifestations short after the vaccination tend to spontaneously subside, which suggests that they do not associate with major changes in disease pathophysiology and do not need major escalations in patient therapy. Therefore, we decided to rely on the concept of sustained activity and persistence of symptoms to support a definition of proper disease flare, mirroring what usually happens in routine rheumatology practice besides vaccination. This consideration has now been highlighted in the methods. Mechanistic considerations have been also expanded in the Discussion.

In the discussion paragraph, there is a lot of rewriting of the content of the RESULT. Please remove these and add a little more discussion content.

  • Answer

We modified the conclusions. According to other Reviewers’ comments we modified the discussion adding some topics.

Round 2

Reviewer 2 Report

The authors have adequately addressed all of my concerns.  Most importantly, the addition of historical control data significantly strengthens the manuscript.  

Reviewer 3 Report

Last time I requested more discussion on the higher rate of flares after vaccination in berimumab user. Berimumab suppresses B lymphocyte maturation, but does not directly suppress T lymphocyte activity. It is possible that the higher rate of post-vaccination flares in berimumab users is due to the activation of the cells which are not suppressed by berimumab.
Although various possibilities should be considered, the authors argue that this phenomenon is caused by the limited application of this drug. (Berimumab is only available for severe state patients.) Are other possibilities intentionally excluded?
I wonder the discussion is not enough, but this paper contains a lot of interesting data. In the context of a global pandemic, some may wait for these information. Although it is an unavoidable decision, it is judged to be acceptable.